# Receptor-Interacting Serine/Threonine-Protein Kinase-2 as a Potential Prognostic Factor in Colorectal Cancer

**DOI:** 10.3390/medicina57070709

**Published:** 2021-07-14

**Authors:** Rola F. Jaafar, Zeid Ibrahim, Karim Ataya, Joelle Hassanieh, Natasha Ard, Walid Faraj

**Affiliations:** 1Department of Surgery, American University of Beirut Medical Center, Beirut 1107 2020, Lebanon; rj29@aub.edu.lb (R.F.J.); iz13@aub.edu.lb (Z.I.); jh91@aub.edu.lb (J.H.); 2Division of Liver Transplantation, Hepatobiliary and Pancreatic Surgery, Department of General Surgery, American University of Beirut Medical Centre, Beirut 1107 2020, Lebanon; ka71@aub.edu.lb; 3Department of General Medicine, Faculty of Medicine, American University of Beirut Medical Center, Beirut 1107 2020, Lebanon; noa09@aub.edu.lb

**Keywords:** colorectal cancer, inflammatory pathways, NFKB, RIPK2

## Abstract

*Background and objectives*: Receptor-interacting serine/threonine-protein kinase-2 (RIPK2) is an important mediator in different pathways in the immune and inflammatory response system. RIPK2 was also shown to play different roles in different cancer types; however, in colorectal cancer (CRC), its role is not well established. This study aims at identifying the role of RIPK2 in CRC progression and survival. *Materials and methods*: Data of patients and mRNA protein expression level of genes associated with CRC (RIPK2, tumor necrosis factor (TNF), TRAF1, TRAF7, KLF6, interlukin-6 (Il6), interlukin-8 (Il8), vascular-endothelial growth factor A (VEGFA), MKI67, TP53, nuclear factor-kappa B (NFKB), NFKB2, BCL2, XIAP, and RELA) were downloaded from the PrognoScan online public database. Patients were divided between low and high RIPK2 expression and different CRC characteristics were studied between the two groups. Survival curves were evaluated using a Kaplan–Meier estimator. The Pearson correlation was used to study the correlation between RIPK2 and the other factors. Statistical analysis was carried out using SPSS version 25.0. The Human Protein Atlas was also used for the relationship between RIPK2 expression in CRC tissues and survival. Differences were considered statistically significant at *p* < 0.05. *Results*: A total of 520 patients were downloaded from the PrognoScan database, and RIPK2 was found to correlate with MKI67, TRAF1, KLF6, TNF, Il6, Il8, VEGFA, NFKB2, BCL2, and RELA. High expression of RIPK2 was associated with high expression of VEGFA (*p* < 0.01) and increased mortality (*p* < 0.01). *Conclusions*: In this study, RIPK2 is shown to be a potential prognostic factor in CRC; however, more studies are needed to assess and verify its potential role as a prognostic marker and in targeted therapy.

## 1. Introduction

The role of inflammation in promoting cancer cell progression is currently a well-known phenomenon [1,2]. Cancer tissues show signs of inflammation, such as the presence of immune cells in the tissue, presence of specific chemokines, and angiogenesis [3]. Chronic inflammation causes tissue damage, which induces cell proliferation and tissue repair and, as a consequence, tumor development [4,5,6].

Receptor-interacting serine/threonine-protein kinase-2 (RIPK2) is an important mediator required in different pathways in the immune and inflammatory response system, and was found to be involved in different solid tumors [7,8]. It is highly expressed in head and neck squamous cell carcinoma (HNSCC), and it was reported to promote cell proliferation and prevent apoptosis in glioma [9,10]. In breast cancer, mainly in triple-negative breast cancer (TNBC), it was shown to impact patient overall survival, increase recurrence, protect cells from apoptosis induced by chemotherapy, and enhance cell proliferation by activating nuclear factor-kappa B (NFKB) [11,12,13]. RIP2 expression correlated with the tumor size, metastasis, overall staging, progression-free survival, and body mass index (BMI) of patients with breast cancer, and RIPK2 polymorphism was also involved in the development of bladder cancer [14,15]. In addition, RIPK2 overexpression is associated with cell proliferation and progression of gastric cancer, and is frequently amplified in lethal prostate cancers, leading to disease progression and aggressiveness [16,17].

Inflammation is known to be an essential tumorigenic factor in colorectal cancer (CRC), and several markers have been suggested to play a role in CRC mediation [18]. Strong evidence suggests that NFKB-mediated inflammation is a key element in the etiology of CRC [19]. RIPK2 is associated with the NFKB pathway and seems to have a role in colitis-associated CRC, where the level of expression of RIPK2 was significantly higher in the colonic mucosa of patients with ulcerative colitis compared to controls [20]. In mice, a deficiency in RIPK2 can cause dysbiosis, which is a microbial imbalance in the colon, which in turn predisposes mice to communicable colitis and colitis-associated CRC [21]. In a recent study on four patients with CRC, RIPK2 was shown to be upregulated in rectal cancer in comparison to normal adjacent mucosa, as identified by the ChIP-Seq procedure [22].

Moreover, RIPK2 expression was reported to be associated with the expression of proto-oncogenic proteins including proliferation marker KI67 (MKI67), tumor protein P53 (TP53), and vascular endothelial growth factor A (VEGFA) [14,15]. These proteins also play a role in the survival and prognosis of CRC patients, where high MKI67 expression is correlated with decreased overall survival and disease-free survival, TP53 expression was found to be significantly associated with poor survival, and VEGF expression was associated with decreased survival, higher grade, presence of lymph node metastasis, depth of invasion, and overall stage [23,24,25,26,27,28].

As mentioned, RIPK2 overexpression is associated with several solid tumors acting through activation of the NFKB and inflammatory pathways. Therefore, this study aims at identifying the role of RIPK2 in CRC patients’ prognosis and survival and its association with the different pro-survival proteins involved in CRC, presenting RIPK2 as a potential biomarker and a therapeutic target.

## 2. Materials and Methods

mRNA proteins’ expression in CRC were downloaded from PrognoScan online public database [29]. mRNA expression level of genes associated with CRC tumorigenesis including RIPK2 and defined proto-oncogenic proteins MKI67, TP53, and VEGF, in addition to tumor necrosis factor (TNF), TNF receptor-associated factor 1 (TRAF1), TRAF7, kruppel-like factor 6 (KLF6), interleukin-6 (Il6), interleukin-8 (Il8), NFKB, NFKB2, B-cell lymphoma 2 (Bcl2), X-linked inhibitor of apoptosis protein (XIAP), and v-rel reticuloendotheliosis viral oncogene homolog A (RELA) were analyzed. Data available included patients’ age, gender, follow-up time, CRC site, histological markers including grade, and TNM staging [30]. Data were downloaded and entered into Statistical Package for the Social Sciences (SPSS) version 25.0 for analysis.

Data was analyzed based on RIPK2 expression as low or high relative to the mean. RIPK2 expression in each dataset was normally distributed. After removal of outliers, RIPK2 mean was determined in each database as a cut-off value and RIPK2 expression above the mean was considered as high, and expression below the mean was considered as low. Additionally, the expression of MKI67, TP53, and VEGF was normally distributed, and low and high expression were defined based on the mean.

Datasets were combined to analyze the association between low and high expression of RIPK2 and different CRC characteristics. Survival curves were evaluated using the Kaplan–Meier estimators. Pearson correlation was used in each dataset separately to study the correlation between RIPK2 and the other proteins. Differences were considered statistically significant at *p* < 0.05.

## 3. Results

A total of four databases with a total of 520 patients were obtained from the PrognoScan database and are detailed in Table 1.

### 3.1. RIPK2 Expression Is Associated with Tumor Site and Grade

Comparing patients based on RIPK2 expression, 273 (52.5%) patients were found to have low expression and 247 (47.5%) had high expression (Table 2). The mean age was 65.54 ± 12.40 and 64.96 ± 13.77, respectively. Higher RIPK2 expression was observed in 16 (59.2%) patients with colon cancer and 11 (40.7%) patients with rectal cancer (*p* = 0.08). Higher expression of RIPK2 was also associated with an increased proportion of grade 3 tumors in 36 patients (27.7%) compared to lower expression in 25 patients (17.2%) (*p* = 0.09). There was no statistically significant association between RIPK2 expression and lymph node involvement, metastasis, and overall stage (Table 2).

### 3.2. RIPK2 Association with Proto-Oncogenes

An association between RIPK2 and other proto-oncogenic proteins’ mRNA expression was obtained (Table 3). The RIPK2 association with high MKI67 mRNA expression was moderate (*p* = 0.06) and significant with high VEGFA mRNA expression (*p* < 0.01), while no significant association was found with TP53 expression.

A correlation analysis was also carried out to understand the relationship between the expression of all the basic proteins mentioned above and RIPK2 expression (Table 4). MKI67, TRAF1, KLF6, Il6, Il8, VEGFA, and RELA were found to positively correlate with RIPK2 and the highest correlation was found between RIPK2 and RELA in the GSE12945 dataset (k = 0.43; *p* < 0.01). TNF and BcL2 mRNA expression negatively correlated with RIPK2 in one dataset each with *p* < 0.01. However, for NFKB2, it was shown to positively correlate with RIPK2 in the GSE12945 dataset (k = 0.42, *p* < 0.01), but a negative correlation was found in another dataset, GSE17537 (k = −0.30, *p* ≤ 0.05). TP53, NFKB, XIAP, and TRAF7 mRNA expression were not significantly correlated with RIPK2 mRNA expression in any of the datasets.

### 3.3. Effect of RIPK2 on the Survival of CRC Patients

Patients with higher RIPK2 mRNA expression had a significantly higher mortality rate (94 patients, 38.06%) in comparison to those with relatively lower expression (61 patients, 22.34%) (*p* < 0.01). Survival analysis was also carried out based on RIPK2 mRNA expression, where high expression of RIPK2 was associated with decreased survival in CRC patients (Figure 1).

Sub-analysis of two databases (GSE12945 and GSE1433) reporting tumor grade and stage was carried out based on survival and is detailed in Table 5. Survival outcome was associated with tumor stage and RIPK2 expression. Moreover, RIPK2 expression was associated with tumor stage, where CRC stage 3 and 4 had significantly more RIPK2 expression relative to stages 1 and 2 (Figure 2).

## 4. Discussion

CRC is the third most common cancer and accounts for 10% of all annually diagnosed cancer, and 9% of cancer-related deaths [31]. Prognostic factors of CRC include the TNM staging, based on which therapeutic decisions are made, and many potential molecular prognostic markers have been described; unfortunately, most have very limited value in routine clinical practice [32].

Surgery, chemotherapy, radiotherapy, and targeted therapy are all options in metastatic CRC. Target therapy includes anti-angiogenetic factors such as bevacizumab, anti-epidermal growth factor receptor-like cetuximab and panitumumab, immune checkpoint inhibitors, anti-BRAF therapy, and HER2-targeted therapy [33]. However, the use of these targeted therapies is limited to factors unique to each patient; for example, the effectiveness of cetuximab is limited to patients with KRAS wild-type tumors, and recent studies also showed that the side of the primary tumor affects the outcome of treatment with cetuximab, with the left-side location being more favorable [34,35]. Hence, discovering other prognostic factors, possible to be targeted by therapy, can improve the outcome of CRC.

We found that the level of mRNA expression of RIPK2 significantly correlated with several proteins involved in tumorigenesis, and after dividing the patients between high and low expression, those who had a higher expression of RIPK2 also had a higher expression of VEGFA (*p* < 0.01). In addition, single database analysis showed a positive correlation between RIPK2 expression and different markers that are established to have a role in CRC, mainly Il-6, Il-8, and VEGF (Table 4), which indicates a potential involvement of RIPK2 in driving tumorigenesis. However, this correlation was not consistent among the different databases, which might be due to the number of patient samples or the data type. Hence, further studies are needed to establish this association. Higher expression of RIPK2 was also associated with worse survival (*p* < 0.01), which suggests RIPK2 as a potential prognostic marker in CRC.

Exploring the Human Protein Atlas, immunohistochemistry analysis was carried out for patients with colon cancer and those with rectal cancer, even though no results were significant, but the patients with rectal cancer were the only ones who showed results similar to our analysis, in which patients with higher expression of RIPK2 showed more death (data not shown). From the data collected from PrognoScan, only 62 patients (11.92%) had information about the site of cancer, so we were not able to carry out separate analysis by site, but with the results shown in the Human Protein Atlas, we can assume that the majority of our patients had rectal cancer.

Grade and the presence of metastasis are important prognostic factors in CRC, and in our analysis based on the PrognoScan data, RIPK2 expression was associated neither with grade (*p* = 0.09) nor with the presence of metastasis (*p* = 0.25), but it was associated with long-term survival (*p* < 0.01). This is basically due to the nature of the data and the availability of tumor grade in only two datasets, which also limited our ability to perform multivariate analysis. However, when analyzing the datasets reporting tumor stage (GSE12945 and GSE1433), a significant association was observed between high RIPK2 expression and tumor grade, which indicates that RIPK2 is a potential prognostic marker, but more large studies and databases reporting tumor characteristics are needed. However, another limit to our study is that information regarding grade and the presence of metastasis was only present for 275 (52.88%) and 61 (11.73%) patients, respectively. Furthermore, no data was available describing the therapy regimen taken by the patients, and hence the association between RIPK2 and survival based on therapy cannot be reported in these datasets. However, some studies showed that RIPK2 plays a role in metastasis in different cancer forms; for example, in TNBC, RIPK2 knockdown decreases migration and lung metastasis, in inflammatory breast cancer, higher RIPK2 activity was correlated with metastasis, and in hepatic cell carcinoma, knockdown of RIPK2 downregulated multiple genes involved in epithelial–mesenchymal transition [12,14,36].

Moreover, data concerning the relationship between RIPK2 and recurrence of CRC was not available in the studied datasets. It was shown that in TNBC, higher expression of RIPK2 is associated with increased recurrence; hence, further studies should be conducted to determine the role of RIPK2 in metastasis and recurrence in CRC, to see if it differs depending on the site of the tumor, and to assess its status as a potential target for therapy in metastatic CRC [13]. So far there is no targeted therapy for RIPK2 in cancer in general and, hence, in CRC; however, several RIPK2 inhibitors have been under investigation in several inflammatory diseases [37]. However, RIPK2 inhibitors are promising therapeutic drugs for cancer, especially inflammation-associated cancer. Therefore, further studies identifying RIPK2 as a prognostic factor and tumor marker in CRC entails a potential hope for CRC targeted therapy.

## Figures and Tables

**Figure 1 medicina-57-00709-f001:**
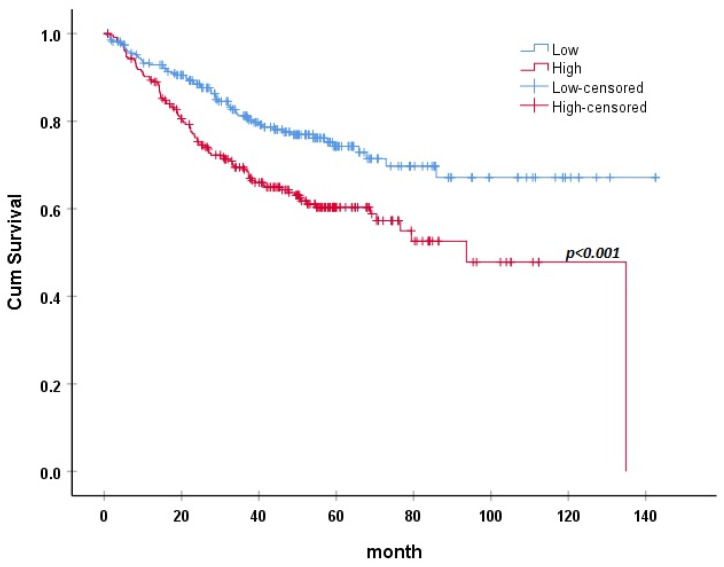
Kaplan–Meier survival plot of patients based on RIPK2 expression.

**Figure 2 medicina-57-00709-f002:**
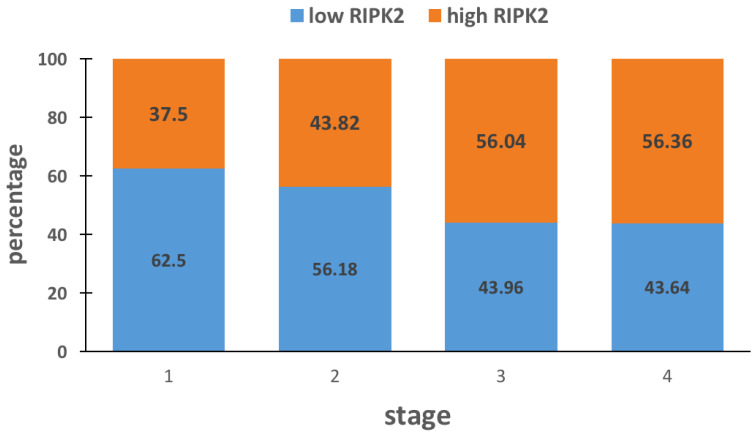
RIPK2 expression relative to tumor stage.

**Table 1 medicina-57-00709-t001:** CRC PrognoScan dataset characteristics.

Dataset	Cohort	Contributor	Year	*n*	Array Type	Age (Mean ± SD)
GSE12945	Berlin	Staub	2009	62	HG-U133A	64.45 ± 11.78
GSE17536	MCC	Smith	2009	177	HG-U133_Plus_2	65.48 ± 13.08
GSE1433	Melbourne	Jorissen	2010	226	HG-U133_Plus_2	66.03 ± 13.01
GSE17537	VMC	Smith	2009	55	HG-U133_Plus_2	62.31 ± 14.35

**Table 2 medicina-57-00709-t002:** Patients’ characteristics according to RIPK2 expression level.

Variable	Low Expression of RIPK2 (*n* = 273)	High Expression of RIPK2 (*n* = 247)	*p*
Age (mean ± SD)	65.54 ± 12.40	64.96 ± 13.77	0.61
Sex			0.85
Male	146 (53.4%)	130 (52.6%)	
Female	127 (46.6%)	117 (47.4%)	
Site			0.08
Colon	13 (37.1%)	16 (59.3%)	
Rectum	22 (62.9%)	11 (40.7%)	
Tumor grade			0.09
1	11 (7.6%)	6 (4.6%)	
2	109 (75.2%)	88 (67.7%)	
3	25 (17.2%)	36 (27.7%)	
T			0.41
2	9 (25.7%)	7 (25.9%)	
3	25 (71.4%)	17 (62.9%)	
4	1 (2.8%)	3 (11.1%)	
N			0.50
0	22 (62.8%)	14 (51.8%)	
1	6 (17.1%)	8 (29.6%)	
2	7 (20.0%)	5 (18.5%)	
M			0.25
0	30 (88.2%)	26 (96.3%)	
1	4 (11.8%)	1 (3.7%)	
Stage			0.19
I	51 (18.7%)	31 (12.5%)	
II	100 (36.6%)	89 (36.0%)	
III	90 (32.9%)	98 (39.7%)	
IV	32 (11.7%)	29 (11.7%)	

**Table 3 medicina-57-00709-t003:** Association of RIPK2 with proliferation genes’ expression.

High Expression of Gene	Low Expression of RIPK2 (*n* = 273)	High Expression of RIPK2 (*n* = 247)	*p*
**MKI67**	139 (47.2%)	141 (57.0%)	0.06
**TP53**	166 (60.8%)	155 (62.7%)	0.65
**VEGFA**	116 (42.4%)	139 (56.2%)	<0.01

**Table 4 medicina-57-00709-t004:** Correlation between RIPK2 mRNA expression and proteins involved in colorectal cancer.

Dataset	MKI67	TRAF1	KLF6	TNF	Il6	Il8	VEGFA	NFKB2	BCL2	RELA
GSE12945	0.16	0.15	0.13	0.09	0.02	0.25 *	0.18	0.42 **	0.22	0.43 **
GSE17536	0.10	−0.03	0.24 **	0.12	0.22 **	0.41 **	0.19 *	−0.06	−0.27 **	0.12
GSE1433	0.15 *	0.16 *	0.20 **	−0.00	0.25 **	0.04	0.20 **	0.12	0.01	0.15 **
GSE17537	0.40 **	0.00	0.12	−0.35 **	0.12	−0.08	−0.00	−0.30 *	−0.19	−0.01

* *p* ≤ 0.05; ** *p* < 0.01.

**Table 5 medicina-57-00709-t005:** Sub-analysis of two databases reporting tumor stage and grade.

Variable	Event (*n* = 99)	No Event (*n* = 176)	*p*
Age (mean ± SD)	65.98 ± 13.98	64.48 ± 12.04	0.35
Sex			0.95
Male	53 (53.0%)	96 (53.3%)	
Female	47 (47.0%)	84 (46.7%)	
Site			0.80
Colon	6 (50.0%)	23 (46.0%)	
Rectum	6 (50.0%)	27 (54.0%)	
Tumor grade			0.20
1	4 (4.0%)	13 (7.4%)	
2	68 (68.7%)	129 (73.3%)	
3	27 (19.3%)	34 (27.3%)	
Stage			<0.01
I	5 (5.0%)	36 (20.0%)	
II	17 (17.0%)	73 (40.6%)	
III	33 (33.0%)	60 (33.3%)	
IV	45 (45.0%)	11 (6.1%)	
TP53	61 (61.0%)	110 (61.1%)	0.98
VEGF-a	54 (54.0%)	77 (42.8%)	0.07
MKI67	51 (51.0%)	93 (51.7%)	0.91
RIPK2	57 (57.0%)	75 (41.7%)	0.014

## Data Availability

The data that support the findings of this study are available from the corresponding author upon reasonable request.

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
