# Peer review of "Receptor-Interacting Serine/Threonine-Protein Kinase-2 as a Potential Prognostic Factor in Colorectal Cancer"

_medicina, 2021, doi:10.3390/medicina57070709_

Round 1

Reviewer 1 Report

The authors present the results of a study aimed at exploring the role of RIPK2 as a potential prognostic marker for colorectal cancer. While the quest for new prognostic markers and potential treatment targets for personalized medicine should be encouraged, I have some concerns regarding the methodology behind this particular study. 

Major concerns:

  1. while the authors state that their selection of this particular marker is based on the link between inflammation and CRC development in ulcerative colitis since RIPK2 is involved in inflammatory pathways, it is unclear whether the analyzed patient population is an adequate reflection of this hypothesis (although not stated, I would assume only a minority of analyzed cases in the PrognoScan database are IBD-related cancers). I recommend a better explanation of why this particular marker was chosen and a more thorough argument in favor of the chosen methodology
  2. it is unclear whether stratifying the groups  in high vs low expression of RIPK2 based on MEDIAN values of expressed protein levels is adequate. is there any previous data to suggest an adequate cut-off level for RIPK2 expression? If not, better explanation of the selected threshold is required, including, at a minimum, a representation of RIPK2 levels in the PrognoScan database and discussion of distribution (normal, non-normal distribution, outliers etc). This is a major issue which determines the validity of all subsequent
  3. comparing mortality rates based on the RIPK2 marker should probably account for a variety of potential confounders (age of patient, stage of disease, clinical status, therapy etc) - so a multivariate analysis is warranted before RIPK2 can be proposed as a potential prognostic marker. it certainly seems a more in-depth analysis is warranted, since there is seems to be no difference in RIPK2 expression according to tumor stage as stated in the results section, and this is, of course, the major drive for disease prognosis and patient outcome. the authors should reconcile this apparent contradiction
  4. the relevance of identifying RIPK2 as a potential prognostic marker should be better discussed - are there any potential therapeutic targets envisioned? this needs to be discussed in the introduction or discussion section, to argue for the relevance of RIPK2 as a marker in CRC management

Author Response

Response to reviewer 1:

Many thanks for your extensive and helpful comments regarding our paper.

  1. while the authors state that their selection of this particular marker is based on the link between inflammation and CRC development in ulcerative colitis since RIPK2 is involved in inflammatory pathways, it is unclear whether the analyzed patient population is an adequate reflection of this hypothesis (although not stated, I would assume only a minority of analyzed cases in the PrognoScan database are IBD-related cancers). I recommend a better explanation of why this particular marker was chosen and a more thorough argument in favor of the chosen methodology

It is unfortunate that the Prognoscan database doesn’t specify the CRC as IBD-related or not, and hence is a limitation. However, our analysis identified that RIPK2 is potential in CRC and further studies are needed to specify its effect in CRC types. In addition, the choice of the marker is based on the link between RIPK2 and IBD, and the association of RIPK2 expression with other solid tumors such as gastric, breast and prostate (highlighted in introduction). Therefore, we assessed its association in publically available datasets so that we can have little insights so that we proceed for well-designed clinical and translational studies

  1. it is unclear whether stratifying the groups in high vs low expression of RIPK2 based on MEDIAN values of expressed protein levels is adequate. is there any previous data to suggest an adequate cut-off level for RIPK2 expression? If not, better explanation of the selected threshold is required, including, at a minimum, a representation of RIPK2 levels in the PrognoScan database and discussion of distribution (normal, non-normal distribution, outliers etc). This is a major issue which determines the validity of all subsequent

We apologize for the mistake; the analysis was based on MEAN not Median, despite the fact that the mean and median were equal in the different datasets (table below). We identified the mean of gene expression in each database, and mean was our cut-off point. Distribution was close to normal in all datasets, where the mean of RIPK2 expression across the databases was close to median and mode and outliers were removed from analysis.

database

median

mean

mode

Gse1294

6.730

6.725

6.739

Gse17536

9.427

9.420

9.430

Gse17537

9.377

9.349

9.260

Gse1433

8.536

8.520

8.561

  1. comparing mortality rates based on the RIPK2 marker should probably account for a variety of potential confounders (age of patient, stage of disease, clinical status, therapy etc) - so a multivariate analysis is warranted before RIPK2 can be proposed as a potential prognostic marker. it certainly seems a more in-depth analysis is warranted, since there is seems to be no difference in RIPK2 expression according to tumor stage as stated in the results section, and this is, of course, the major drive for disease prognosis and patient outcome. the authors should reconcile this apparent contradiction

As reported in table 2, there were no difference in age, gender, tumor grade or stage between patients with high vs low RIPK2 expression, so the multivariate analysis would not affect outcome. However, additional analysis based on mortality is outlined in Table 5.  

Regarding the difference between RIPK2 expression and tumor stage as drive for disease prognosis, this is really a very valid point. However, the main limitation of this analysis is the availability of tumor stage and grade data. Only one database (GSE12945, 62 patients) describe the tumor stage and grade. Therefore, the statistical analysis is sufficient.

However, this is the first analysis that sheds light on the potential role of RIPK2 in CRC, and further studies to validate its role are needed.

  1. the relevance of identifying RIPK2 as a potential prognostic marker should be better discussed - are there any potential therapeutic targets envisioned? this needs to be discussed in the introduction or discussion section, to argue for the relevance of RIPK2 as a marker in CRC management

There are FDA drugs approved that can specifically target RIPK2. This is added to discussion section

Reviewer 2 Report

Summary

Colorectal cancer (CRC) is the third most common cancer in the world.  In this research paper, the authors investigated the potential role of receptor interacting-serine/threonine-protein-kinase -2 (RIPK2) in CRC progression and survival.  Using the PrognoScan online public database, the data from patients with regard to mRNA protein expression levels of genes known to be associated with CRC such as RIPK2, TNF, TRAF1, TRAF7, KLF6, IL-6, IL8, VEGF-A, MK167, TP53, NF-KB, NFKB2, BCL2, XIAP and RELA were compared.  Survival curves were evaluated using a Kaplan-Meier estimator, Pearson correlation was used to study correlation between RIPK2 and other factors, and human protein atlas was also used for the relation between RIPK2 expression in CRC tissues and survival.  They observed that RIPK2 expression levels were correlated with MK167, TRAF1, KLF6, TNF, IL-6, IL-8, VEGFA, NFKB2, BCL2 and RELA.  Importantly, high expression levels of RIPK2 was associated with high expression of VEGFA, and increased mortality.  The authors conclude that RIPK2 is a potential prognostic and therapeutic marker in CRC.      

Minor comments:

  1. Please revise the introduction, please explain clearly the role of RIPK2 in the development of cancer.

Author Response

Many thanks for your review and comment. We have edited the introduction and added the required information and added more recent references regarding the role of RIPK2 in solid tumor development. RIPK2 has been implicated in several cancers. Recent studies regarding its role in tumor progression and aggressiveness have been reported including breast, gastric, prostate.. However, this is the first study on its role in CRC. Definitely, further studies are required 

Round 2

Reviewer 1 Report

Although the authors tried to address the previous round of comments, there are 2 major flaws in the study design which limit the applicability of the findings.

  1. the link between IBD (an area where RIPK2 has a clearly established role) and colon cancer in this particular cohort is very difficult to argue - since it is unclear how many (if any) IBD-related CRC have been included in the cohort. Although hypothetically there could be a link between the two processes, the study design does not fully support this link
  2. the authors do not provide any significant evidence to show that, even if RIPK2 is relevant to CRC, there is no data in the refence section to support the potential clinical relevance of this finding => reference 37 which was inserted after the first round of comments does not show any evidence of targeted drugs FOR CRC, rather it highlights the potential of RIPK2-directed drugs in IBD 
